# MMKG-PAR: Multi-Modal Knowledge Graphs-Based Personalized Attraction Recommendation

**Gengyue Zhang** [1,2], **Hao Li** [1,2,*], **Shuangling Li** [1,2], **Beibei Wang** [1,2] and **Zhixing Ding** [1,2]

1    Yunnan Key Laboratory of Digital Communications, Kunming 650504, China;
     zhanggengyue@stu.ynu.edu.cn (G.Z.); lishuangling@itc.ynu.edu.cn (S.L.); wang316_95@163.com (B.W.);
     dzxdwd@mail.ynu.edu.cn (Z.D.)
2    Intelligent Tourism Engineering Research Center, School of Information Science & Engineering,
     Yunnan University, Kunming 650091, China
*    Correspondence: lihao707@ynu.edu.cn

**Abstract:** As the tourism industry rapidly develops, providing personalized attraction recommendations has become a hot research area. Knowledge graphs, with their rich semantic information and entity relationships, not only enhance the accuracy and personalization of recommendation systems but also energize the sustainable development of the tourism industry. Current research mainly focuses on single-modal knowledge modeling, limiting the in-depth understanding of complex entity characteristics and relationships. To address this challenge, this paper proposes a multi-modal knowledge graphs-based personalized attraction recommendation (MMKG-PAR) model. We utilized data from the "Travel Yunnan" app, along with users' historical interaction data, to construct a collaborative multi-modal knowledge graph for Yunnan tourist attractions, which includes various forms such as images and text. Then, we employed advanced feature extraction methods to extract useful features from multi-modal data (images and text), and these were used as entity attributes to enhance the representation of entity nodes. To more effectively process graph-structured data and capture the complex relationships between nodes, our model incorporated graph neural networks and introduced an attention mechanism for mining and inferring higher-order information about entities. Additionally, MMKG-PAR introduced a dynamic time-weighted strategy for representing users, effectively capturing and precisely describing the dynamics of user behavior. Experimental results demonstrate that MMKG-PAR surpasses existing methods in personalized recommendations, providing significant support for the continuous development and innovation in the tourism industry.

**Keywords:** personalized recommendation; multi-modal knowledge graph; graph attention mechanism; user representation strategy; sustainable development

## 1. Introduction

In tourism planning and development, sustainability is a key consideration. Integrating advanced artificial intelligence technology can effectively address the environmental, economic, and social challenges faced by the tourism industry, thereby achieving the goals of sustainable tourism. Additionally, providing personalized attraction recommendations to tourists demonstrates its value on multiple levels [1]. For tourists, such recommendations can substantially enhance the overall travel experience, meeting a variety of personalized needs and interests. Additionally, personalized recommendations play a crucial role in broader domains, including the sustainable development of the tourism industry, tourism destination marketing, and the prosperity of local economies [2]. However, as the number of travel options continues to grow, tourists often face a common challenge: how to make wise choices among numerous attractions and activities to ensure a travel experience that is not only enriching but also meets personalized needs?

Traditional recommendation algorithms are mainly divided into three categories: content-based recommendations [3], collaborative filtering (CF)-based recommendations [4,5], and hy-

brid recommendations [6]. Content-based recommendation relies on matching project features, which can lead to overly consistent results, lacking diversity. Collaborative filtering-based recommendations, which analyze similarities between users or items, often face challenges with new users or items, a problem commonly referred to as the cold start issue. Hybrid recommendations combine the advantages of content-based and collaborative filtering-based methods, but they are complex to implement and may still be affected by the individual limitations of each algorithm [7]. Overall, these traditional recommendation algorithms perform poorly in terms of diversity in personalized needs and data sparsity, especially in tourism recommendation systems where these issues may be more pronounced.

Knowledge graphs (KGs) are structured methods of data representation that effectively capture and organize diverse information and their complex interactions in the real world through nodes (i.e., entities) and edges (i.e., relationships) [8]. In the context of tourist attraction KGs, entities include attractions and their labels (e.g., type and city), while the relationships between these entities (e.g., located in) are considered edges. For instance, the triple (Erhai Lake, attraction.city, Dali) in Figure 1 indicates that Erhai Lake is located in the city of Dali. In the field of attraction recommendation, KGs significantly enhance the accuracy and interpretability of recommendation systems by providing rich structured semantic information. Compared to traditional content-based and collaborative filtering recommendation algorithms, recommendation systems based on KGs are more effective in addressing the cold start problem [8–10]. They are capable of leveraging the relationships and attribute [8,11] information between entities to provide highly relevant and high-quality recommendations, even in the absence of user interaction data. Furthermore, KGs offer a more comprehensive understanding and representation of user needs and attraction characteristics. By analyzing various aspects related to a particular attraction, such as type, climate, and geography, KGs can deliver richer and more personalized recommendations to tourists. However, it is important to note that structured KGs in a single data dimension might not fully represent entities, as they typically capture information from a specific dimension (e.g., text or image). This can lead to limitations in the recommendations under certain circumstances.

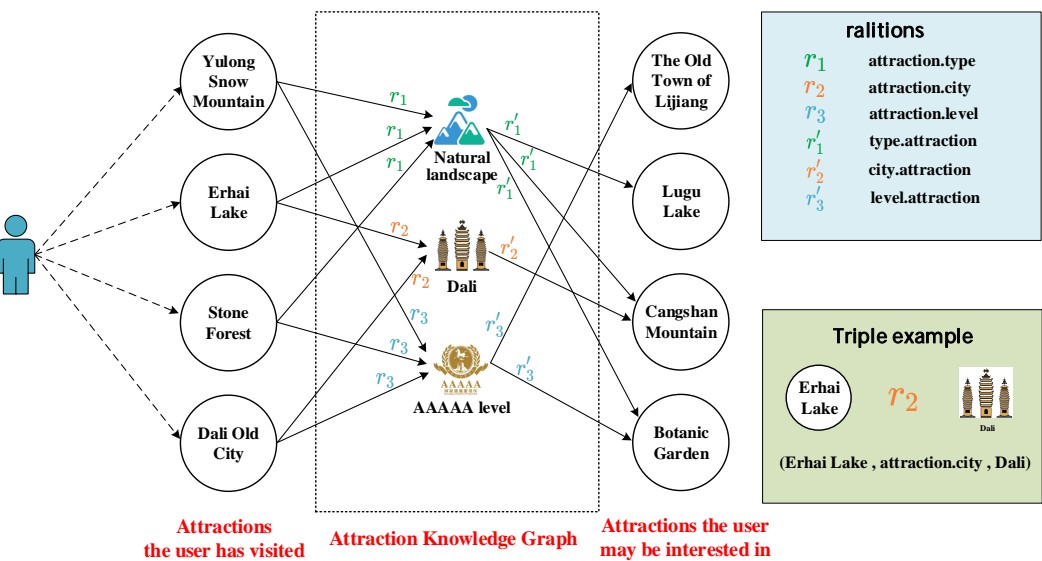

**Figure 1.** Illustration of knowledge graph-enhanced tourist attraction recommendation system.

Recent studies have shown that user preferences are often influenced by multi-modal information [12,13], particularly through multi-modal representations of symbols and signs (as exemplified by the case of Prague) [14]. This further substantiates that, within the tourism sector, the integration of images and textual information not only enriches the

source of information but also enhances the accuracy and reliability of recommendation systems by providing a more comprehensive background and cultural context. Consequently, researchers have begun incorporating these pieces of information into the recommendation frameworks of KGs. Textual information, such as historical backgrounds, cultural contexts, and tourist reviews about tourist attractions, can offer users in-depth insights, assisting them in making wiser choices. A specific attraction may gain favor from specific user groups due to its rich history or unique cultural activities, and this information is typically fully presented only in textual descriptions. Meanwhile, image information provides a more intuitive visual experience, showcasing the visual appeal of attractions and their related activities or facilities, such as nearby hotels or restaurants. By integrating textual and image information, multi-modal knowledge graphs (MMKGs) [15,16] are expected not only to show significant effectiveness in tourist attraction recommendations but also to support the sustainable development of the tourism industry. This approach will enhance destination marketing, providing the tourism sector with an effective promotional strategy.

In response to the description above, this paper integrates the recommendation process with MMKGs, proposing a personalized tourist attraction recommendation model based on MMKGs named MMKG-PAR. Specifically, MMKG-PAR models the MMKGs of tourist attractions in the following aspects:

- The model employs a time-weighted user embedding strategy, aggregating users' historical interaction information into the collaborative multi-modal knowledge graph (CMKG, detailed in Section 3.3). This approach captures users' changing interests over time more precisely, significantly enhancing the accuracy and personalization of the recommendation system.
- The model utilizes advanced feature extractors for multi-modal data, effectively combining features from different modalities to enhance entity representation in the knowledge graph.
- The model utilizes the message propagation mechanism of graph neural networks (GNNs) [17,18], recursively aggregating information from adjacent nodes to update the feature representation of each node. In this process, by incorporating an attention mechanism [19], it learns the weights of each neighboring node during propagation, thereby enabling the extraction and induction of higher-order information.

The MMKG-PAR model was evaluated on two tourist attraction CMKG datasets. In these datasets, MMKG-PAR's Recall@20 improved by 6.67% and 3.95%, and NDCG@20 improved by 5.30% and 3.61%, respectively. The main contributions of this paper include are as follows:

- To the best of our knowledge, this is the first application of multi-modal knowledge graph in the field of tourist attraction recommendation. The multi-modal knowledge graph we developed integrates images and textual information of attractions, offering a more comprehensive view of each site. Furthermore, the feature fusion method presented in this paper effectively consolidates multi-modal information and serves as a robust example for integrating additional modal features of attractions.
- The MMKG-PAR model proposed in this paper effectively integrates various modal attribute data to enhance the representation of entity nodes. It employs an aggregation strategy based on graph attention mechanisms for efficiently merging higher-order information, resulting in more precise learning and representation.
- Utilizing real data from the "Travel Yunnan" app, this study constructed two tourist attraction CMKG datasets. The MMKG-PAR was compared with several state-of-the-art methods. Extensive experiments validated the model's rationality and effectiveness.

To aid reader comprehension, we have listed common abbreviations used in this paper in Table 1.

**Table 1.** Summary of the abbreviations involved in this paper.

| Abbreviation | Expansion |
|:---:|:---:|
| KG | Knowledge graph |
| MMKG | Multi-modal knowledge graph |
| CF | Collaborative filtering |
| CMKG | Collaborative multi-modal knowledge graph |
| GNN | Graph neural network |
| PCA | Principal component analysis |
| MKGE | Multi-modal knowledge graph embedding |
| GCN | Graph convolutional network |

## 2. Related Work

This section primarily reviews existing work related to our research, encompassing multi-modal knowledge graphs (MMKGs) and recommendations based on MMKGs.

### 2.1. Multi-Modal Knowledge Graph

MMKGs are forms of knowledge representation that integrate various types of data, such as text and images. By incorporating these diverse data types into MMKGs, the concept of traditional knowledge graphs is expanded, making the expression of knowledge richer and more multidimensional. MMKGs are better equipped to simulate the complexity and diversity of the real world, particularly excelling in scenarios that require the integration of visual and textual information.

Figure 2 demonstrates the two primary forms of representation in MMKGs: "multi-modal knowledge graph—entity type" (MMKG-E) and "multi-modal knowledge graph—attribute type" (MMKG-A) [20]. MMKG-E considers multi-modal information as independent entities, emphasizing the interrelations and interactions among these information elements, suitable for applications in multimedia content analysis and cross-media information retrieval. Conversely, MMKG-A treats multi-modal information as attributes of existing entities, focusing on describing the multifaceted characteristics of individual entities, making it well-suited for applications in personalized recommendation systems and the construction of datasets for deep learning training.

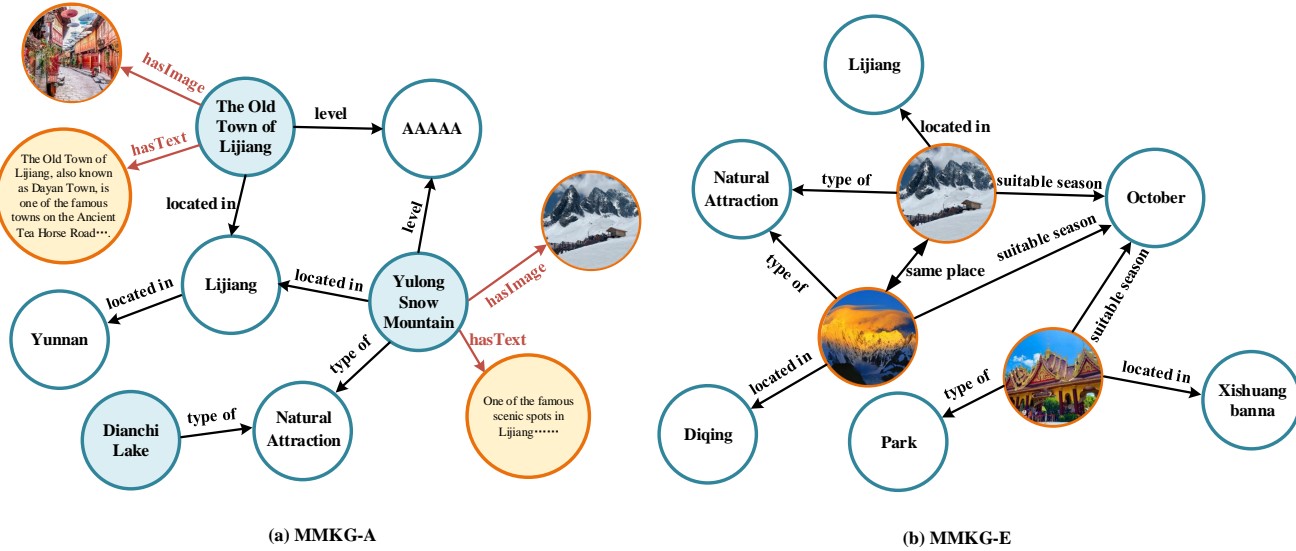

**Figure 2.** Illustration of multi-modal knowledge graphs: (**a**) in MMKG-A, multi-modal information is treated as attributes of entities; (**b**) in MMKG-E, multi-modal information is considered as independent entities.

This paper employs the MMKG-A approach, which treats the multi-modal information of tourist attractions as attributes of the attraction entities, thereby achieving a deep integration of information. This combination of structured knowledge, text, and visual information in a multi-modal knowledge graph presents a more comprehensive and multi-dimensional model of attraction information, enriching and enlivening the representation of these attractions. This approach, aligning with the standards of high-level artificial intelligence research, offers a nuanced and vibrant depiction of tourist spots.

### 2.2. MMKG-Based Recommendation

In the field of artificial intelligence, recommendation systems based on MMKGs are garnering widespread attention. However, research on making recommendations using MMKGs is still relatively limited. These systems enhance accuracy and efficiency by integrating various data modalities such as text, images, and audio. For instance, the MMGCN [21] model, by constructing user-item interaction graphs in each modality and aggregating information using graph convolutional networks (GCNs) [22,23], innovatively introduces visual, auditory, and textual modalities for micro-video recommendations. GRCN [24] builds upon MMGCN, refining the bipartite user-item graph with weighted edges based on user preferences and item content affinity, yet both MMGCN and GRCN fall short in effectively integrating modalities. MMKGR [25] explores multi-hop reasoning in the knowledge graph domain using a unified gate-attention network with multi-modal auxiliary features. MKGAT [26] applies multi-modal graph attention technology in MMKG-E, representing multi-modal information like text and images as distinct entity features, enhancing information dissemination and recommendation system performance, albeit with limitations in simulating complex inter-modal relations. MMKDGAT [27] integrates attributes and visual information of remote sensing images, enriching node representations with multi-modal information and high-order collaborative signals but exhibits a relatively singular modality application.

It is noteworthy that, in the aforementioned methods, users and entities within a multi-modal knowledge graph are treated as equivalent nodes. However, this approach may not fully capture the essential differences between users and entities: users are dynamic with individualized needs and preferences, while entities are typically static with fixed attributes. Treating both as equivalent nodes might not adequately reflect these differences, potentially impacting the performance of the recommendation systems to some extent.

## 3. Problem Formulation

This section initially introduces a set of preliminary concepts, followed by the formulation of a task recommendation based on MMKGs.

### 3.1. User-Attraction Interaction Graph

When recommending tourist attractions, it is common to rely on historical interactions between users and attractions, (e.g., clicks and browsing activities). Here, we define the set of users as $\mathcal{U} = \{u_1, u_2, u_3, \ldots\ldots, u_M\}$ and the set of attractions as $\mathcal{I} = \{i_1, i_2, i_3, \ldots\ldots, i_N\}$, where $M$ and $N$ represent the total number of users and attractions, respectively. The historical interactions between users and attractions are structured into a user-attraction interaction graph, represented as $\mathcal{G}_1 = \{(u, i) \mid u \in \mathcal{U}, i \in \mathcal{I}\}$. An edge exists between a user $u$ and an attraction $i$, marked as $y_{ui} = 1$, if there has been interaction; if there is no record of interaction, then $y_{ui} = 0$.

### 3.2. Multi-Modal Attraction Knowledge Graph

As shown in Figure 2a, the multi-modal attraction knowledge graph we constructed incorporates multi-modal information as inherent attributes of the entities. We define the multi-modal tourist attraction knowledge graph as $\mathcal{G}_2 = \{\mathcal{E}, \mathcal{R}, \mathcal{A}, T_\mathcal{A}, T_\mathcal{E}\}$, where $\mathcal{E}$, $\mathcal{R}$, and $\mathcal{A}$, respectively, represent sets of entities, relations, and attributes. $T_\mathcal{A}$ and $T_\mathcal{E}$ denote attribute triples and entity triples, respectively. $T_\mathcal{A}$, formalized as $\{(e, r, a) \mid e \in \mathcal{E}, r \in \mathcal{R}, a \in \mathcal{A}\}$, is

used to represent attributes of an attraction entity. For example, the triple (Jade Dragon Snow Mountain, hasImage, attraction image) indicates that the attraction entity has an image attribute, describing some visual information of the attraction. Similarly, $T_{\mathcal{E}}$, formalized as $\{(e_1, r_1, e_2) \mid e_1, e_2 \in \mathcal{E}, r_1 \in \mathcal{R}\}$, represents the relationships between entities. For instance, the triple (Jade Dragon Snow Mountain, located in, Lijiang) describes the city where the attraction is situated. This structure effectively captures the complex relationships and attributes within the dataset.

### 3.3. Collaborative Multi-Modal Knowledge Graph (CMKG)

Herein, we propose the CMKG, seamlessly integrating user-attraction interaction graphs with multi-modal attraction knowledge graphs through item connections into a unified CMKG. This integrated structure enhances the interactive data and provides a comprehensive informational view for users, supporting more accurate recommendations and decision-making. Considering the attraction set $\mathcal{I}$ as a subset of the entity set $\mathcal{E}$, we define the attraction–entity alignment set as $\mathcal{Z} = \{(i, e) \mid i \in \mathcal{I}, e \in \mathcal{E}\}$, where $(i, e)$ indicates that an attraction $i$ can align with an entity $e$ within the knowledge graph. We define the collaborative multi-modal knowledge graph as $\mathcal{G} = \{(u, \textit{Interact}, i) \mid u \in \mathcal{U}, (i, e) \in \mathcal{Z}, (u, t_{ui}, i) \in \mathcal{G}_1\}$, and, as illustrated in Figure 3, an example of a CMKG is provided.

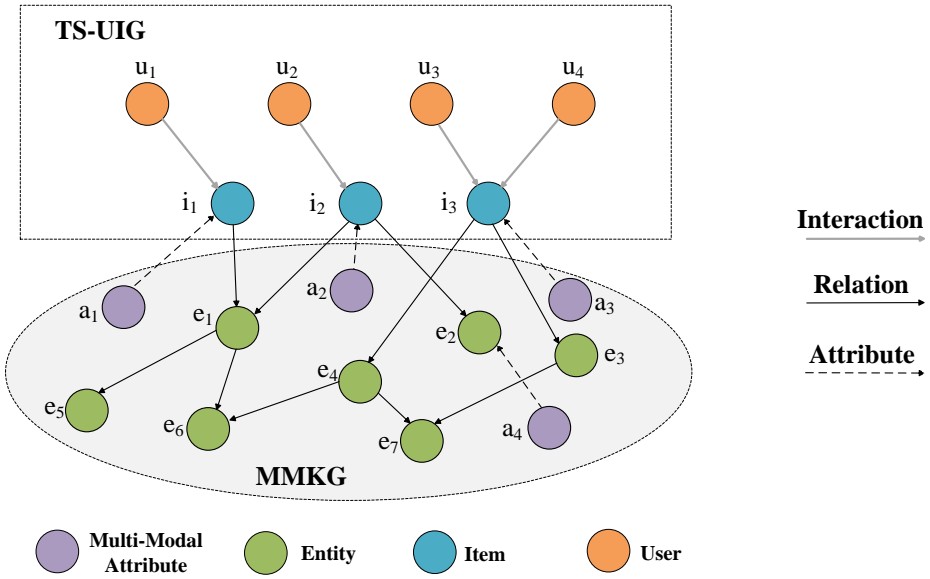

**Figure 3.** Illustration of a collaborative multi-modal knowledge graph.

### 3.4. Task Description

Currently, we have devised a personalized tourist attraction recommendation task based on MMKGs as follows:

- **Input:** the collaborative multi-modal knowledge graph $\mathcal{G}$, comprising the user-attraction interaction graph $\mathcal{G}_1$ and the multi-modal attraction knowledge graph $\mathcal{G}_2$.

- **Output:** a predictive function designed to estimate the probability of interaction between a user $u$ and a previously uninteracted attraction $i$, denoted as

$$\hat{y}_{ui} = \mathcal{F}(u, i \mid \mathcal{G}, \Theta), \tag{1}$$

where $\hat{y}_{ui}$ represents the probability of recommending attraction $i$ to user $u$, and $\Theta$ denotes the model parameters.

## 4. Method

In this section, we introduce our proposed MMKG-PAR model, which employs a novel user representation strategy incorporating a dynamic time-weighted mechanism to innovatively capture the temporal dynamics of user behavior. The model also utilizes advanced feature extraction methods to derive useful features from multi-modal data (e.g., images, text), enhancing entity node representations with these attributes. To effectively process graph-structured data and capture complex inter-node relationships, our model incorporates GNNs and introduces an attention mechanism for mining and summarizing high-order entity information. Figure 4 illustrates the overall framework of the MMKG-PAR model, which is primarily composed of three parts: (1) CMKG embedding layer, (2) propagation layer, and (3) prediction layer. A detailed introduction to each of these components will be provided in the following sections.

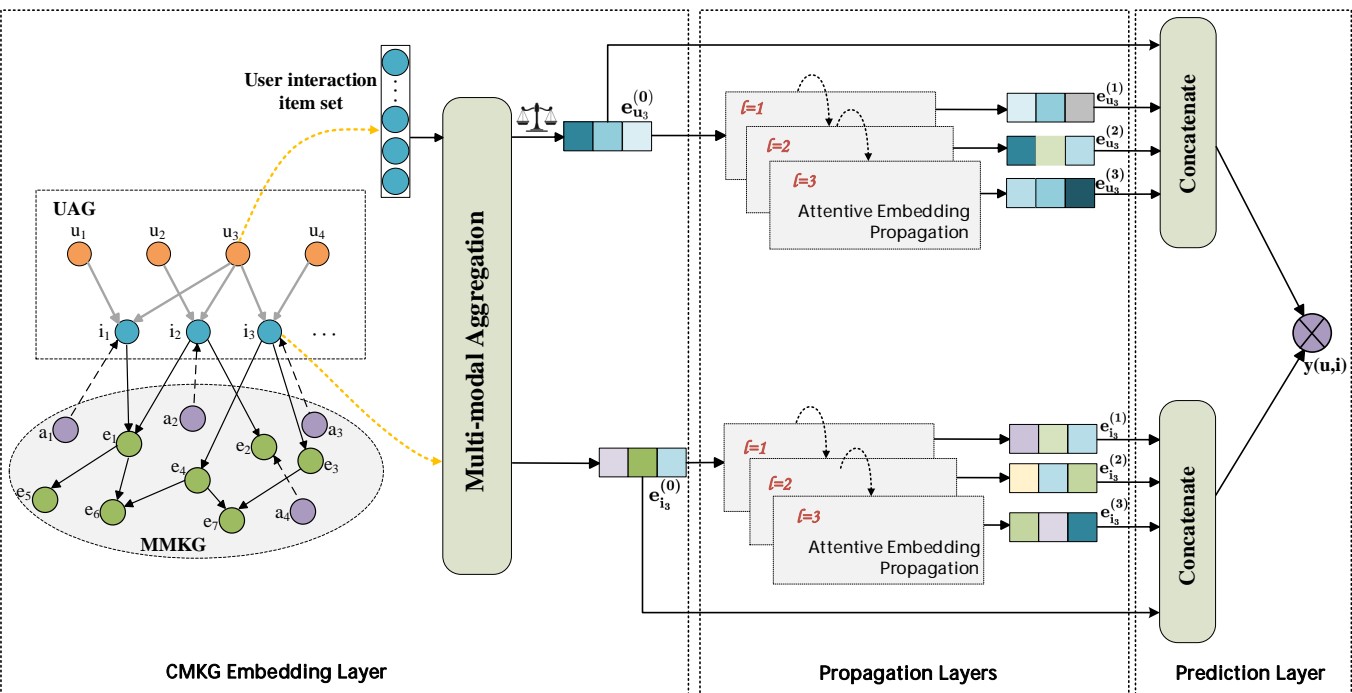

**Figure 4.** Illustration of the MMKG-PAR model's overall framework. The left side features the CMKG embedding layer, the central part comprises the propagation layer, and the right side consists of the prediction layer.

### 4.1. CMKG Embedding Layer

As we have discussed in Sections 3.3 and 3.4, the CMKG is used as the input for the MMKG-PAR model.

### 4.1.1. User Embedding

In this subsection, we will thoroughly introduce and discuss the user embedding strategies that we have implemented. In addition to traditional direct user encoding methods, our research also incorporates two innovative user embedding approaches.

- **Direct user embedding method:** This is a conventional user embedding strategy where each user is directly encoded as a unique embedding vector $e_u$. This approach is intuitive and straightforward, with user embeddings typically learned through the model, reflecting individual user characteristics. To maintain consistency in representation, the symbol $e_u$ will also be used to denote user embeddings in subsequent discussions of different user representation methods.
- **Interaction-averaged embedding method:** In this strategy, the embedding of a user is calculated as the arithmetic mean of the embedding vectors of all items they have

interacted with. Suppose the set of attractions a user $u$ has interacted with is denoted as $I_u = \{i_1, i_2, \ldots, i_n\}$, where each attraction $e_{ik}$ has a corresponding embedding $e_{ik}$. The embedding $e_u$ of user $u$ can be obtained by calculating the arithmetic mean of the embeddings of all attractions they have interacted with, as illustrated below:

$$e_u = \frac{1}{n} \sum_{k=1}^{n} e_{ik}. \tag{2}$$

- **Interaction-time-weighted embedding method:** Similar to the average embedding method, this approach incorporates a time-weighting mechanism for calculating user embeddings. Interactions closer to the current time are assigned greater weight, allowing the embedding vector to more significantly reflect the user's recent interests and preferences. Assuming that the weight of each interaction is determined by a function $w(k)$, where the value of $w(k)$ decreases as the time interval between the interaction and the current time increases, the embedding vector e for user u can be represented by the following formula:

$$e_u = \frac{\sum_{k=1}^{n} w(k) \cdot e_{ik}}{\sum_{k=1}^{n} w(k)}, \tag{3}$$

where $w(k)$ is the weight assigned based on the temporal distance between the interaction and the current time. In this study, we opted to define the weight using an exponential decay function, denoted as $w(k) = e^{-\lambda \cdot t_k}$. Here, $\lambda$ represents the decay rate, and $t_k$ is the time elapsed from the occurrence of the interaction to the current time. The choice of this exponential decay form was based on the assumption that a user's more recent interactions have a greater influence on their current preferences than earlier ones. Through this weighted strategy, we are able to more accurately capture the dynamics of user preferences over time, thereby providing user embeddings that are more closely aligned with the user's current interests for prediction and recommendation systems.

### 4.1.2. Multi-Modal Data Fusion

In this subsection, we conduct an embedding analysis based on the structural features of the multi-modal knowledge graph for smart tourism in Yunnan Province. Our method enhances the representation of entities by utilizing multi-modal data as attributes. The multi-modal fusion strategy is depicted in Figure 5. Next, we provide a detailed introduction to the processing methods for each type of modal data, as well as the fusion strategy:

- **Structured entity:** In the process of handling structured entities in knowledge graphs, we assigned a unique identifier (ID) to each entity. These IDs were transformed into dense vector representations, namely entity embeddings $e_s$, via an embedding layer. The embedding layer is responsible for learning the mapping function $f : \text{ID} \rightarrow \mathbb{R}^d$, where $d$ represents the dimension of the embedding space.

- **Image data:** We utilized the deep residual network ResNet-50 [28] for extracting image features in our research. This network, pre-trained on the ImageNet dataset, is adept at capturing high-level semantic information from images. Subsequently, the visual features extracted using ResNet-50 were subjected to dimensionality reduction through principal component analysis (PCA) [29]. These features were then nonlinearly mapped using the fully connected layer $FC_{img}$, with the aim of projecting the visual data into the embedding space of structured entities. The processing of the image feature vector is represented as follows:

$$e_{vis} = ReLU(FC_{img}(PCA(\text{ResNet-50}(Image)))), \tag{4}$$

where $ReLU$ represents a nonlinear activation function, ensuring the effective integration of visual features with entity embeddings within the same dimensional space.

This process, by optimizing the representation of visual features and incorporating them into entity embeddings, enhances the semantic representation of entities in the knowledge graph, providing visual support for multi-modal recommendations.

- **Textual data:** For encoding textual information, we selected BERT [30] (Bidirectional Encoder Representations from Transformers) as our text encoder to thoroughly extract rich knowledge associated with entities. Proposed by Devlin et al., the BERT model has set new benchmarks in various NLP tasks due to its superior language representation capabilities. We utilized the pre-trained BERT-Base [30] model, specifically its pooled output, to generate descriptive embeddings for each entity, resulting in a 768-dimensional textual feature vector. In integrating textual features, we also employed PCA to reduce the dimensionality of the high-dimensional features outputted by BERT, optimizing computational efficiency and enhancing the scalability of the model. Ultimately, the textual features were mapped through a fully connected layer, $FC_{text}$, ensuring alignment with the feature space of entity embeddings. The processing of textual features is represented as

$$e_{txt} = ReLU(FC_{text}(PCA(\text{BERT-Base}(Text)))). \tag{5}$$

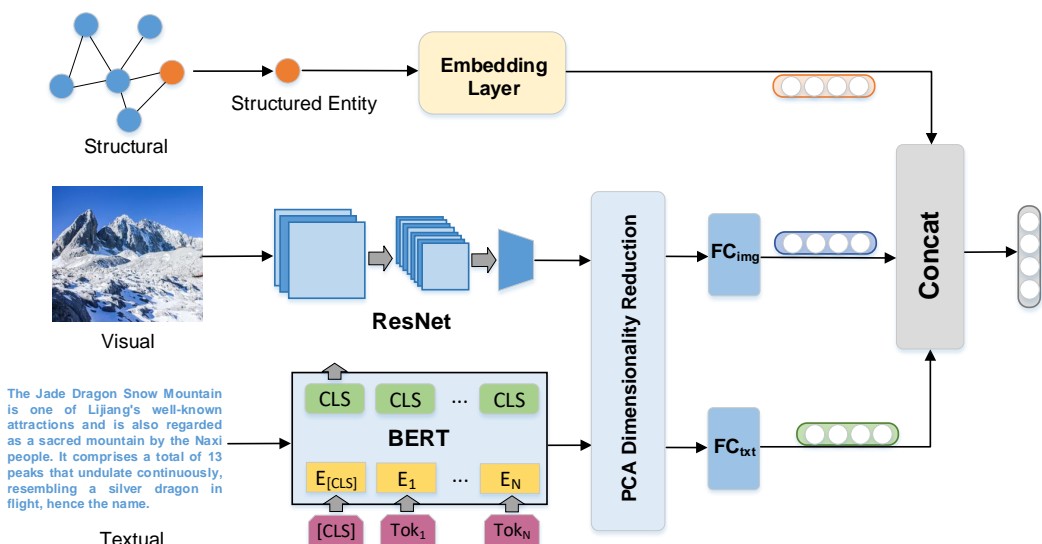

**Figure 5.** An example of the multi-modal data fusion process. Image and textual data are used as attributes of entities to enhance entity representation.

To fully leverage multi-modal information for enhancing the semantic representation of entities in knowledge graphs, this study employed three multi-modal fusion strategies. These strategies not only consider the structured features of the entities but also integrate deep insights from image and textual data, thereby achieving a comprehensive enhancement of entity features:

- **Vector concatenation:** In the vector concatenation strategy, feature vectors from different modalities are initially concatenated together. Subsequently, they are transformed through a fully connected layer to map them into a common feature space. This process is represented as follows:

$$e_{con} = W_{con}(Concat(e_s, e_{vis}, e_{txt})) + b_{con}. \tag{6}$$

- **Vector addition:** In this strategy, the feature vectors of structured entities, text, and images are summed up to balance the contributions of each modality. This combined vector then undergoes a nonlinear transformation. This process is represented as

$$e_{add} = W_{add}(e_s + e_{vis} + e_{txt}) + b_{add}. \tag{7}$$

- **Maximal vector selection:** In this strategy, a single modality is selected, specifically the feature vector that contains the maximum amount of information among all modalities, to represent the overall entity. This process is represented as

$$e_{sel} = W_{sel}(SelectMaxVar(e_s, e_{vis}, e_{txt}) + b_{sel}. \tag{8}$$

Ultimately, the representation of entities enhanced with multi-modal attributes is defined as

$$e_{mul} = W_{mul} \cdot f(e_s, e_{vis}, e_{txt}) + b_{mul}, \tag{9}$$

where $f(e_s, e_{vis}, e_{txt})$ is a high-level feature fusion function that could be any of the aforementioned fusion strategies. The weight matrix $W_{mul}$ and bias term $b_{mul}$ for the chosen fusion strategy ensure that the final output is aligned with the original entity embedding in the same dimensional space. The effectiveness of these three feature fusion strategies in enhancing entity representations, as well as their specific impacts on the performance of downstream tasks, will be examined in subsequent experiments.

4.1.3. Multi-Modal Knowledge Graph Embedding (MKGE)

In the MKGE module, our goal was to represent entities and relations as vectors while preserving the graph's structural information. To achieve this, we used the TransE [31] method, known for its unique advantages in handling entity and relation embeddings. Specifically, for each triple $(h, r, t)$ in the graph, we expected $e_h + e_r \approx e_t$, where $e_h$ and $e_t$ represent entity embeddings, and $e_r$ is the relation embedding. Through this method, we formalize the energy scoring function as follows:

$$S(h, r, t) = \|e_h + e_r - e_t\|^2, \tag{10}$$

where a lower energy score indicates that the triple is more likely to be true. During the training process, we employed a pairwise ranking loss to differentiate between positive examples (i.e., valid triples) and negative examples (i.e., intentionally corrupted triples). The loss function is designed as

$$\mathcal{L}_{MKGE} = -\sum_{(h,r,t,t') \in \mathcal{T}} \log \sigma(S(h, r, t') - S(h, r, t)), \tag{11}$$

where $\mathcal{T}$ includes the valid triple $(h, r, t)$ and the corresponding corrupted triple $(h, r, t')$. By applying the TransE method, the MKGE module not only gains a profound understanding of entities and their relations but also effectively quantifies the complex interactions between entities, providing a solid foundation for further analysis and recommendation tasks.

*4.2. Propagation Layer*

In CMKGs, the role of the information propagation layer is to utilize the GNN's architecture to reveal both direct and indirect connections between tourist attractions. Taking Figure 6 as an example, we consider each tourist attraction entity as a hub of information. These entities enrich their feature representations by exchanging information with connected entities, such as geographical location and historical background. To ensure consistency and efficiency in the computation of each batch, we randomly select a fixed number of neighbors for each entity, rather than using all of their neighbors. During this process, we employ a knowledge-aware attention mechanism to determine the weights of information propagation, ensuring that more significant relationships receive greater focus during the exchange. Ultimately, through the aggregation of information, a comprehensive feature representation for each attraction is constructed, providing a rich informational foundation for subsequent recommendation systems or other analytical tasks. Below, we describe this process in detail, encompassing single-layer propagation, information aggregation, and higher-order propagation.

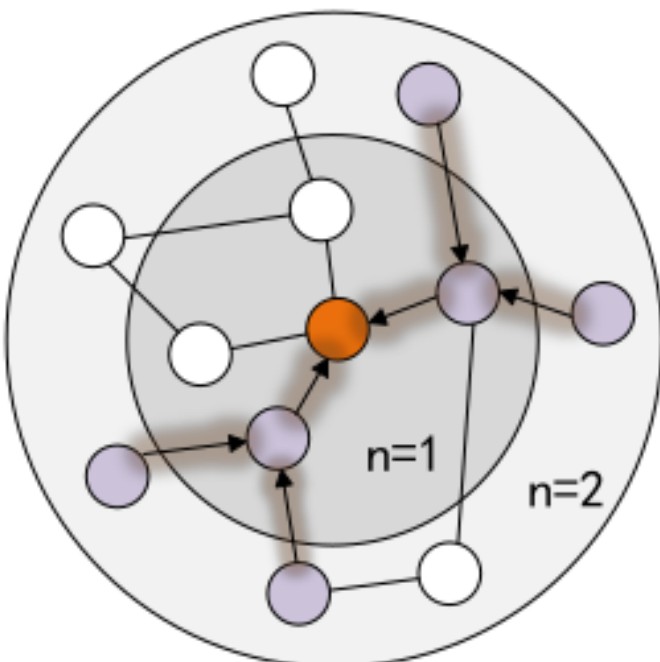

**Figure 6.** Two-hop propagation of the orange entity in GNNs.

### 4.2.1. Single-Layer Propagation

In our multi-modal tourist attraction knowledge graph, we incorporate graph attention networks (GATs) [19] and emphasize the importance of relational embeddings. The representation of entities, such as specific attractions, is updated by aggregating information from adjacent entities. When focusing on how to characterize the relationship between a given entity and its adjacent entity $h$, we adopt a weighted aggregation strategy to compute the information aggregation vector $e_{\mathcal{T}_h}$ for the entity $h$. This vector is a linear combination of the embeddings of all tail entities $t$ in the adjacent triple set $\mathcal{T}_h = \{(h, r, t) | (h, r, t) \in \mathcal{G}\}$ of the entity, represented as

$$e_{\mathcal{T}_h} = \sum_{(h,r,t) \in \mathcal{T}_h} \lambda_{(h,r,t)} e_t, \tag{12}$$

where $e_t$ denotes the embedding vector of the tail entity $t$, and $\lambda_{(h,r,t)}$ denotes a computed attention weight. The process of calculating the attention weight $\lambda_{(h,r,t)}$ involves several steps, starting with the computation of a preliminary attention score $s_{(h,r,t)}$, represented as

$$S_{(h,r,t)} = \text{LeakyReLU}(W_r x_h + W_r x_t + b_r), \tag{13}$$

where $W_r$ is a transformation matrix related to the relationship $r$, and $b_r$ is a bias term. The LeakyReLU function ensures that a gradient flow is maintained, even in the case of negative input values. Subsequently, a softmax function is applied for normalization, determining the final attention weights for each adjacent triple, represented as

$$\lambda_{(h,r,t)} = \frac{\exp(s_{(h,r,t)})}{\sum_{(h,r',t') \in \mathcal{T}_h} \exp(s_{(h,r',t')})}. \tag{14}$$

### 4.2.2. Information Aggregation

After calculating the information aggregation vector $e_{\mathcal{T}_h}$ for the entity $h$, we optimize the aggregation strategy for the entity representation $e_t$ and its corresponding neighborhood representation $e_{\mathcal{T}_h}$. Formally, we define this new type of entity representation as $e'_h = f(e_h, e_{\mathcal{T}_h})$. Specifically, the new entity representation $e'_h$ is generated by the following three aggregators:

- **Sum aggregator:** This aggregator employs a summation strategy to combine two representations, which are further refined through high-order nonlinear transformations. It is represented by the formula

$$agg_{add} = \text{LeakyReLU}(W(e_h + e_{\mathcal{T}_h}) + b), \tag{15}$$

  where $W$ is a trainable weight matrix that maps features from various dimensions to a common space. $b$ is a bias term, providing additional flexibility to the linear transformation. The incorporation of a nonlinear activation function enables the model to capture more complex data patterns. The subsequent two aggregators also utilize the same approach in applying a weight matrix $W$, a bias term $b$, and the LeakyReLU nonlinear activation function; therefore, they are not distinguished further.

- **Concat aggregator:** The concat aggregator employs a strategy of concatenating two representations, initiating with a concatenation followed by complex nonlinear transformations. This approach helps to preserve the information of the original features, while providing a rich input for subsequent linear and nonlinear transformations. The concat aggregator can be represented as

$$agg_{con} = \text{LeakyReLU}(W \cdot \text{concat}(e_h, e_{\mathcal{T}_h}) + b), \tag{16}$$

  where $\text{concat}(e_h, e_{\mathcal{T}_h})$ denotes the concatenation of the embedding vector of the head entity $h$ with the aggregated vector of adjacent entity information.

- **Hybrid feature interaction aggregator:** Inspired by the bi-interaction aggregator in the KGAT [32] model, the hybrid feature interaction aggregator enhances the capability to capture information by combining various types of feature interactions. It is represented as

$$agg_{hyb} = \text{LeakyReLU}((W_1(e_h + e_{\mathcal{T}_h}) + W_2(e_h \odot e_{\mathcal{T}_h}) + b), \tag{17}$$

  where $\odot$ denotes the element-wise product. In the experimental section, we will comprehensively examine the performance of these three aggregators.

### 4.2.3. High-Order Propagation

In exploring the inherent high-order connectivity within collaborative multi-modal knowledge graphs (CMKGs), this paper adopts a multi-layer propagation and aggregation approach. As illustrated in Figure 6, entities can accumulate and utilize neighborhood information from farther distances by stacking more propagation layers. Specifically, for an n-layer model, the captured information will encompass up to the n-th order neighbors. In each propagation layer $l$, an entity's representation is based on its representation from the previous layer and the accumulated information of its neighbors. This process can be formalized by the formula

$$e_h^{(l)} = f\left(e_h^{(l-1)}, H^{(l-1)}(\mathcal{N}_h)\right), \tag{18}$$

where $f$ is a function that integrates the entity's own features with neighborhood information. $H^{(l-1)}(\mathcal{N}_h)$ is an aggregation function that collects and consolidates information from the $l-1$ order neighborhood of $h$ with its computation formula as

$$H^{(l-1)}(\mathcal{N}_h) = \sum_{(h,r,t) \in \mathcal{N}_h} \pi^{(l-1)}(h,r,t)e_t^{(l-1)}, \tag{19}$$

where $\pi^{(l-1)}(h,r,t)$ is a weighting function that determines the contribution of each neighbor at the $l-1$ level, and $e_t^{(l-1)}$ is the representation of entity $t$ generated in the previous propagation step. Although higher-order propagation provides the ability to capture complex neighborhood information, it can also introduce issues such as over-smoothing,

overfitting, and increased computational complexity. The specific number of propagation layers used in the model and their impact on performance and efficiency will be discussed in detail in the subsequent experimental section, ensuring the optimization of the model structure for achieving optimal performance.

*4.3. Prediction Layer*

After the multi-layer propagation process, in order to construct the final representations of user *u* and attraction *i*, we have employed a concatenation operation to integrate information from various layers. The advantage of this concatenation method lies in its ability to preserve the unique information provided by each layer, thereby preventing the loss of information and forming a rich and comprehensive feature representation. This can be expressed as

$$e_u^* = \bigoplus_{l=1}^{L} e_u^{(l)}, \quad e_i^* = \bigoplus_{l=1}^{L} e_i^{(l)}, \tag{20}$$

where $e_u^{(l)}$ and $e_i^{(l)}$, respectively, represent the representations of user *u* and attraction *i* at the *l*-th layer, and $\bigoplus$ denotes the concatenation of these representations along a specific dimension. Based on these composite representations, we further predict the interaction or matching degree between users and attractions by calculating the inner product of their representation vectors:

$$\hat{y}_{ui} = (\mathrm{e}_u^*)^\top \mathrm{e}_i^*. \tag{21}$$

The optimization recommendation strategy employed by MMKG-PAR combines BPR [33] loss and regularization terms. The model utilizes BPR loss to focus on enhancing the prediction accuracy of the matching degree between users and attractions. Concurrently, the introduced regularization terms effectively reduce the risk of overfitting. The specific expression for the loss function is as follows:

$$\mathcal{L}_{Rec} = - \sum_{(u,i,j)\in\mathcal{D}} \log \sigma \big(\hat{y}_{ui} - \hat{y}_{uj}\big) + \lambda \|\Theta\|_2^2, \tag{22}$$

where $\mathcal{D}$ denotes the training dataset composed of user *u*, the attractions *i* with which the user has interacted (i.e., positive samples), and the attractions *j* with which the user has not interacted (i.e., negative samples). $\sigma$ is the sigmoid function, which is utilized to convert the score differences between users and attractions into probability values. The variables $\hat{y}_{ui}$ and $\hat{y}_{uj}$, respectively, represent the predicted scores of user *u* for attractions *i* and *j*. The variable $\lambda$ is the regularization coefficient. The term $\|\Theta\|_2^2$ denotes the L2 norm of the model parameters $\Theta$, used for regularization to prevent overfitting.

**5. Experiments**

*5.1. Experimental Settings*

5.1.1. Datasets Description

Currently, there is a lack of publicly available multi-modal knowledge graph datasets in the field of tourism. In light of this, to comprehensively evaluate the performance of our proposed model, we constructed two simulated datasets based on personalized preferences and conducted a series of exhaustive experiments. These datasets primarily relied on real data provided by the "Travel Yunnan" app, supplemented by the application of web crawler technology, aiming to further enrich and perfect the content of the knowledge graph and ensure that each attraction entity had multi-modal data. In terms of the historical interaction between users and attractions, we extracted information from the 2022 order data of the "Travel Yunnan" app, obtaining about one million interaction records with Yunnan Province attractions from approximately 200,000 users. The two datasets are named MT-6M and MT-13M, respectively. MT-6M contains the full set of data, while MT-13M focuses on about 50,000 users with more than seven interactions in their history, comprising about 660,000 interaction records. Since there is an overlap between these two datasets,

they share a multi-modal tourism knowledge graph. Detailed statistical data are shown in Table 2.

**Table 2.** Basic Statistics of the Two Datasets.

|  | **MT-6M** | **MT-13M** |
|---|---|---|
| #users | 49,935 | 240,115 |
| #items | 1218 | 1218 |
| #interactions | 660,912 | 1,321,054 |
| #entities | 32,184 | 32,184 |
| #relations | 9 | 9 |
| #triples | 362,527 | 362,527 |

5.1.2. Baselines

To demonstrate its effectiveness, we compared our proposed MMKG-PAR model with several of the latest baselines, including methods based on collaborative filtering (CF), KG-based approaches, and methods using MMKGs:

- **BPRMF** [34] is a widely used collaborative filtering technique in recommender systems, specifically designed for implicit feedback data. Its core idea revolves around providing a personalized ranking for each user by leveraging a pairwise ranking loss function.
- **CKE** [35] integrates collaborative filtering and knowledge graph embedding, optimizing the system by combining user interaction data and knowledge graphs to enhance recommendation accuracy through the merging of user behavior and structured knowledge.
- **RippleNet** [10] enhances user representations by integrating regularization and path-based methods. The model places particular emphasis on including relevant item information within user paths to comprehensively capture user interests and preferences.
- **KGAT** [32] combines knowledge graphs with recommendation systems, employing attention mechanisms to process entities and relationships within knowledge graphs. This approach enables more precise learning of the complex interactions between users and items, enhancing the efficacy of recommendation systems in academic research.
- **MMGCN** [21] captures interaction relationships by constructing bipartite graphs of users and items for each modality, and then employs GCNs to train these graphs.
- **MMKDGAT** [27] integrates visual information as entities within a knowledge graph and employs GCNs to aggregate neighbors, thereby enhancing the representation of these entities.

5.1.3. Evaluation Protocols and Parameter Settings

We implemented the MMKG-PAR in PyTorch 2.1 and employed a systematic approach for evaluation and parameter tuning. For each dataset, we randomly assigned 80% of user interactions to the training set and reserved the remaining 20% for the test set. Additionally, 10% of interactions from the training set were selected as a validation set to fine-tune hyperparameters. Interactions between each user and tourist attractions were treated as positive samples, while negative samples were generated by pairing users with attractions they had not previously interacted with. For each user in the test set, MMKG-PAR predicts their preference scores for all tourist attractions, excluding those with positive interactions in the training set. To evaluate the effectiveness of MMKG-PAR, we utilized two widely used metrics: Recall@$K$ and NDCG@$K$, with K set by default to 20.

In terms of model training, we initialized parameters using the Xavier [36] initializer and optimized the model with the Adam [37] optimizer. We explored a variety of hyperparameters, drawing on the optimal settings reported in the original baseline papers, including a batch size of 2048, a regularization coefficient set to $10^{-4}$, and a learning rate of 0.05. To construct a model with third-order connectivity, we set the depth of MMKG-PAR

to three layers. For handling visual attributes, we utilized the 2048-dimensional features from the last hidden layer of ResNet. For textual attributes, we extracted the vector corresponding to the [CLS] token from the last layer of the Transformer [38] network in the BERT-Base model to obtain a 768-dimensional comprehensive feature representation of the text. Finally, we set the feature size d of all entities to 64 dimensions by default for ease of subsequent processing.

### 5.2. Overall Performance Comparison

This section provides a detailed comparative analysis of the performance of MMKG-PAR with baseline models. Table 3 presents the comparative results of all models, from which the following conclusions can be drawn:

- In the MT-6M and MT-13M datasets, the MMKG-PAR model outperforms other models on two key evaluation metrics: Recall@20 and NDCG@20. Notably, compared with the strongest baseline model, on the MT-6M dataset, MMKG-PAR achieves an improvement of 6.67% in Recall@20 and 5.30% in NDCG@20.
- BPRMF, as a collaborative filtering model, performs less effectively compared to other models that incorporate knowledge graph information due to its lack of utilization of additional knowledge graph data. KGAT, which combines GCNs and attention mechanisms, shows improved performance over models that solely utilize knowledge graphs, such as CKE and RippleNet.
- Within the realm of multi-modal knowledge graph-based recommendation methods, the performance of MMGCN and MMKDGAT models indicates that integrating multi-modal data on top of a knowledge graph is an effective means of enhancing recommendation accuracy. Both models surpassed those not utilizing multi-modal data in terms of the Recall@20 and NDCG@20 metrics.
- The performance of MMKG-PAR surpasses that of MMGCN and MMKDGAT, indicating that the multi-modal data fusion strategy employed by MMKG-PAR effectively enhances the representation capability of entities. Consequently, MMKG-PAR demonstrates a more significant improvement in recommendation results.
- The models demonstrate better overall performance in the MT-6M dataset compared to the MT-13M. This could be attributed to the fact that MT-6M, by filtering out users with less than seven interactions and their corresponding data, ensures higher data quality. Such filtering is particularly beneficial for user embedding methods that rely on interaction time weighting, effectively capturing user preferences more accurately.

**Table 3.** Overall performance of recommendations. The asterisk * denotes the baseline model with the strongest performance under the current evaluation metric.

| | MT-6M | | MT-13M | |
|---|---|---|---|---|
| | **Recall@20** | **NDCG@20** | **Recall@20** | **NDCG@20** |
| BPRMF | 0.1157 | 0.0842 | 0.0821 | 0.0730 |
| CKE | 0.1244 | 0.0918 | 0.0960 | 0.0876 |
| RippleNet | 0.1237 | 0.0954 | 0.1019 | 0.0927 |
| KGAT | 0.1331 | 0.1051 | 0.1236 | 0.1028 |
| MMGCN | 0.1353 | 0.1042 | 0.1316 * | 0.1042 |
| MMKDGAT | 0.1411 * | 0.1075 * | 0.1309 | 0.1054 * |
| **MMKG-PAR** | **0.1505** | **0.1132** | **0.1368** | **0.1092** |
| **Improv.%** | 6.67% | 5.30% | 3.95% | 3.61% |

### 5.3. Ablation and Effectiveness Analysis

#### 5.3.1. Effect of Parameters

Figure 7 illustrates the impact of parameter settings on the performance of MMKG-PAR. Concerning embedding dimensions, lower dimensions are insufficient to capture the complexity and subtle differences between entities and relationships, resulting in informa-

tion loss and inferior performance. On the other hand, higher dimensions demand more computational resources and memory, leading to slower training and inference speeds, and potentially causing overfitting. From Figure 7, it is evident that the embedding dimension $d$ has a pronounced influence on the model's performance. Smaller embedding dimensions, such as 16 or 32, fail to adequately represent the intricate relationships between items. When the embedding dimension reaches 128, the Recall@20 and NDCG@20 evaluation metrics reach their peak, indicating that the model's ability to capture information is optimized at this point. However, further increasing the dimension to 256 or 512 results in a decline in performance, possibly due to overfitting, which impairs its generalization capability.

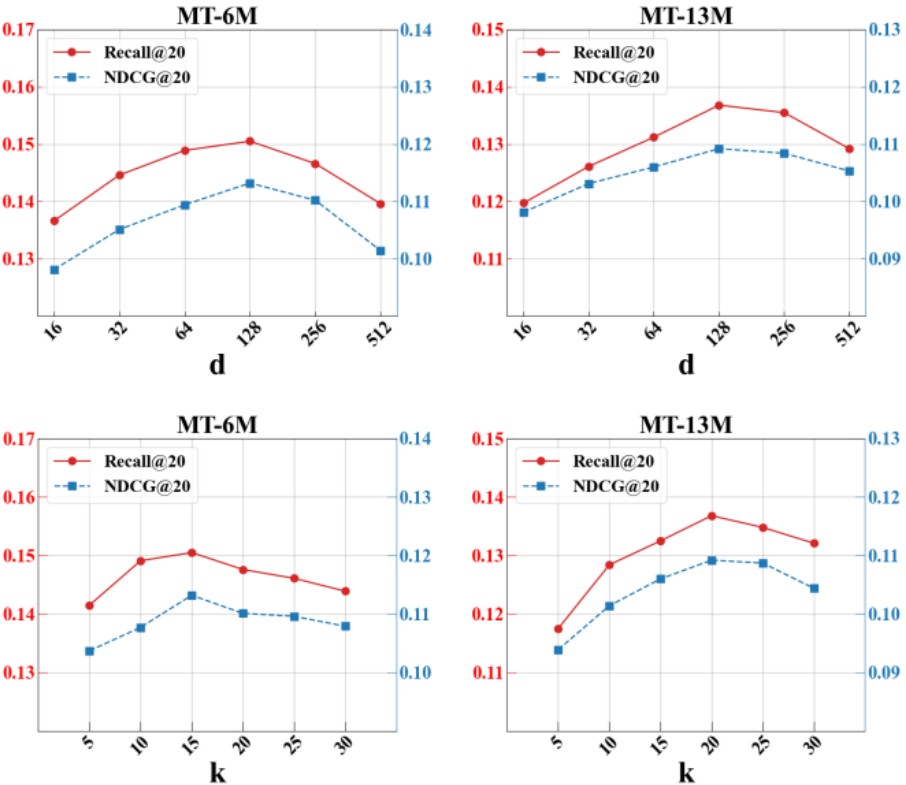

**Figure 7.** Effect of different embedding dimensions $d$ and neighbor counts $k$ on performance.

When studying the impact of the neighbor count $k$ on model performance, we selected only the most similar items to construct an item–item association graph in order to reduce the propagation of unrelated information. As shown in Figure 7, with the increase in the neighbor count $k$, the model's Recall@20 and NDCG@20 metrics on the MT-6M dataset exhibit a trend of initially increasing and then decreasing, reaching their peak performance at $k$ equals 15. In contrast, for the MT-13M dataset, the performance peak occurs at $k$ equals 20. One possible reason for this phenomenon is that the appropriate number of neighbors is crucial for effective information propagation and maintaining computational efficiency. Too few neighbors may result in the loss of critical information, while too many neighbors may introduce noise and increase computational costs. Therefore, finding the optimal balance for the neighbor count is of significant practical importance for improving model recommendation performance and optimizing resource allocation.

5.3.2. Effect of Multi-Modal Information

To investigate the impact of multi-modal data on enhancing entity representations, we constructed various model variants and conducted ablation experiments on the MT-6M and MT-13M datasets, respectively. Table 4 records the performance of these variants on the two datasets, which can be summarized as follows:

- Without any ablations, the baseline model achieved the best performance on both datasets, demonstrating the critical importance of image and text data in enhancing the quality of multi-modal entity representations in knowledge graph recommendation tasks. This impact consistently manifested across datasets of different scales.
- When removing any one modality, the recommendation performance decreased, with the most significant loss occurring when both modalities were removed. However, when considering the results in Tables 3 and 4, even in the absence of multi-modal attributes, MMKG-PAR's performance still surpasses knowledge graph baseline models such as CKE, RippleNet, and KGAT. One possible reason for this is that MMKG-PAR, based on time-weighted user embeddings, better reflects users' latest interests. Additionally, its graph attention mechanism effectively captures neighborhood information, enabling higher-order connectivity and enhancing recommendation accuracy.
- Compared to removing text, the removal of image data had a more significant impact on performance. This could be attributed to the fact that visual information is more likely to capture the user's attention when browsing information about tourist attractions on online platforms. Therefore, visual information is potentially more crucial for recommendation effectiveness than textual information. Additionally, textual descriptions often contain unrelated information, which may confuse users.

**Table 4.** The impact of multi-modal information on recommendation effects.

| | MT-6M | | MT-13M | |
| --- | --- | --- | --- | --- |
| | Recall@20 | NDCG@20 | Recall@20 | NDCG@20 |
| MMKG-PAR | 0.1505 | 0.1132 | 0.1368 | 0.1092 |
| MMKG-PAR w/o Visal | 0.1435 (−4.7%) | 0.1076 (−4.9%) | 0.1324 (−3.2%) | 0.1063 (−2.7%) |
| MMKG-PAR w/o Text | 0.1466 (−2.6%) | 0.1109 (−2.0%) | 0.1347 (−1.5%) | 0.1065 (−2.4%) |
| MMKG-PAR w/o Visal&Text | 0.1416 (−5.9%) | 0.1067 (−5.7%) | 0.1318 (−3.7%) | 0.1047 (−4.1%) |

### 5.3.3. Effect of Model Depth

In this subsection, we delve into the influence of propagation layers within the MMKG-PAR on GNNs. To optimize the utilization of collaborative signals in the dataset while circumventing the introduction of excessive smoothing and noise, a series of experiments were conducted. The findings of these experiments are documented in Table 5. From these experiments, we have derived the following observations:

- In the MT-6M dataset, the model's performance in terms of Recall@20 and NDCG@20 reached its peak when the number of propagation layers in the graph neural network was increased to three. For the MT-13M dataset, the optimal performance for Recall@20 was observed at four layers of propagation, while the best performance for NDCG@20 occurred at three layers. Overall, the trend indicates that the recommendation effectiveness is optimal when the number of propagation layers is between three and four.
- The increase in the number of layers had a dual effect. On the one hand, augmenting the number of layers can emulate more complex interactions by capturing higher-order relationships. On the other hand, an excessive number of layers might introduce noise, leading to stagnation or even a decline in model performance. As demonstrated in Table 5, when the model depth exceeds three layers, there is no significant enhancement in performance and, in some instances, a decrease in performance is observed.

### 5.3.4. Effect of Fusion and Aggregation Methods

Figure 8 illustrates the impact of different user embedding methods, multi-modal data fusion techniques, and aggregation layer strategies in graph neural networks on the

recommendation performance in the MMKG-PAR on two datasets. We fixed the number of propagation layers in the model to one and employed Recall@20 as the evaluation metric.

**Table 5.** The impact of model depth on recommendation performance, where we bold the best performance results.

| | MT-6M | | MT-13M | |
|---|---|---|---|---|
| | Recall@20 | NDCG@20 | Recall@20 | NDCG@20 |
| One layer | 0.1476 | 0.1103 | 0.1346 | 0.1062 |
| Two layer | 0.1497 | 0.1127 | 0.1361 | 0.1087 |
| Three layer | **0.1505** | **0.1132** | 0.1366 | **0.1092** |
| Four layer | 0.1501 | 0.1130 | **0.1368** | 0.1089 |

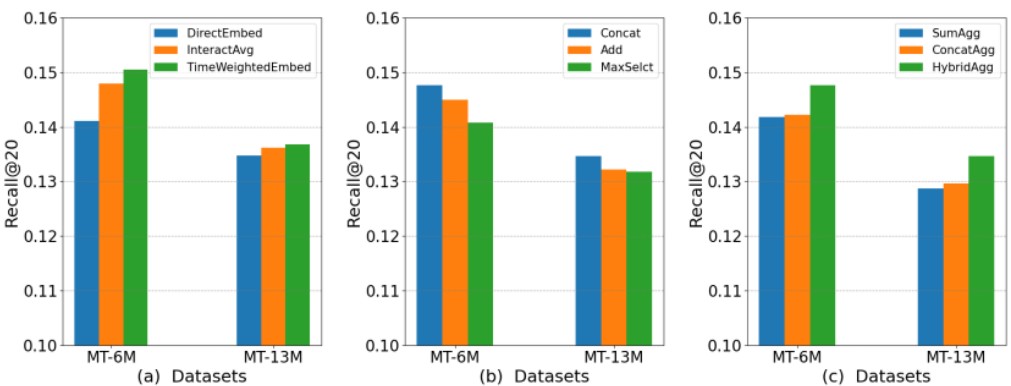

**Figure 8.** Effect of different user embedding methods, multi-modal data fusion techniques, and aggregation layer strategies in GNNs on recommendation performance.

In terms of user embedding strategies, we validated the three different methods mentioned in Section 4.1.1. According to the data in Figure 8a, on the MT-6M dataset, the TimeWeightedEmbed strategy exhibited the best performance. This suggests that considering the interaction time between users and items during the embedding process is beneficial, as it allows the embedding to more accurately reflect changes in user preferences over time, thereby enhancing the accuracy of recommendations. The InteractAvg method derives user embeddings by calculating the arithmetic mean of the embedding vectors of all items a user has interacted with. Compared to DirectEmbed, this method more effectively smooths and synthesizes a user's overall preferences, particularly in dealing with outlier interactions. However, on the MT-13M dataset, the performance advantage of the TimeWeightedEmbed strategy was not significant, which might be due to the insufficiency of relying on a limited number of weighted interactions to reflect user preferences when user interaction data is sparse.

We investigated the three multi-modal data fusion strategies mentioned in Section 4.1.2 through experiments. As shown in Figure 8b, the vector concatenation strategy demonstrated a clear advantage, owing to its preservation of information from all modalities and the application of fully connected layers, enabling the model to learn complex relationships between different modal features. In contrast, the performance of the vector summation strategy was relatively poor. One possible reason is that it simply stacks the feature vectors from different modalities together, potentially leading to the loss of crucial information, especially in cases where the importance of features varies across modalities or when there is complementarity. The maximization vector selection strategy, which chooses the feature vector from a single modality and neglects the information that other modalities might provide, also failed to show significant advantages. This is because all modalities contribute to the entity representation, and no single modality can fully substitute the information from others. This observation is corroborated by the analysis of experimental data in Section 5.3.2.

Figure 8c presents the comparative results of the three aggregators in the graph neural network aggregation layer mentioned in Section 4.2.2. Among these, HybridAgg demonstrated the best performance in terms of recommendation effectiveness. One possible reason is that HybridAgg integrates different aggregation methods and processes interaction features between entities, enabling it to more thoroughly capture the complex relationships within multi-modal data, thereby enhancing the accuracy and relevance of recommendations. Meanwhile, SumAgg and ConcatAgg achieved similar results in terms of recommendation effectiveness, with ConcatAgg having a slight edge. This could be attributed to ConcatAgg's ability to concatenate representations of different entities, more comprehensively preserving the feature information of each modality. This is particularly important when dealing with multi-modal data, thus giving it a slight advantage in performance.

*5.4. Case Study*

To visually demonstrate the impact of multi-modal attribute features, we randomly selected five specific users from the MT-6M dataset and the attractions they are interested in, and visualized them in two dimensions using t-SNE [39]. From the t-SNE bi-dimensional visualization in Figure 9, we can observe the distribution of these five users and their points of interest across structured, visual, textual, and multi-modal feature spaces. By observing the clustering in the multi-modal feature space, it can be found that, compared to single features (e.g., only structure, visual, or textual features), multi-modal features provide a more distinctive representation, causing the attractions of interest to the users to tend to form tighter clusters. This indicates that integrating multi-modal information can better capture user preferences, thereby helping to generate more accurate and personalized recommendations.

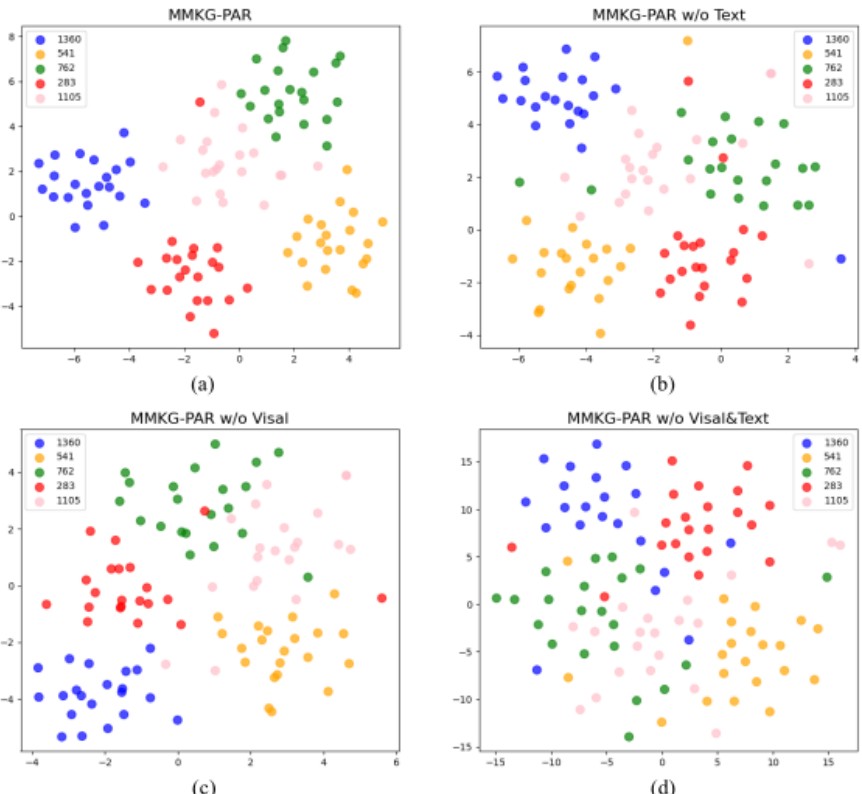

**Figure 9.** Visualization of user preferences in multi-modal feature spaces. Each point represents a user's interested attraction entity, and different colors indicate different users. (**a**) Displays the clustering of data points by the MMKG-PAR model in a two-dimensional space. Meanwhile, (**b**), (**c**), and (**d**) respectively present the distribution of data points when the model excludes text features, visual features, and both visual and text features simultaneously.

## 6. Discussion

Our experimental results demonstrate that the MMKG-PAR model can mine tourists' latent interest preferences through multi-modal knowledge graphs, thereby offering more precise recommendations. We compared the MMKG-PAR model with six baseline models, including BPRMF [34] based on collaborative filtering, CKE [35], RippleNet [10], and KGAT [32] based on KGs, as well as MMGCN [21] and MMKDGAT [27] based on MMKGs. Comparisons with BPRMF confirmed the positive contribution of knowledge graph data to recommendation system performance. The contrast between CKE, RippleNet, and KGAT highlighted the significant role of integrating graph convolutional networks and attention mechanisms in enhancing recommendation accuracy. Finally, comparing MMKG-PAR with MMGCN and MMKDGAT underscored the critical advantages of multi-modal data fusion strategies in improving entity representation and recommendation outcomes.

It is noteworthy that introducing an interaction-time weighted user embedding method significantly enhanced the expressiveness of user embeddings. This approach, by allocating higher weights to recent interactions, enables a more accurate capture of users' evolving preferences over time, thereby offering personalized recommendations that closely align with their current interests and needs. Compared to straightforward averaging or direct embedding methods, the interaction-time-weighted embedding method distinctly excels in capturing the latest trends in user preferences.

At the same time, our study faces some limitations. The absence of publicly available multi-modal knowledge graph datasets in the tourism sector restricts the validation of our model's generalizability. Additionally, our model's dependency on high-quality multi-modal data and the impact of noise during feature extraction increase the complexity of data processing. Given these limitations, our future work will focus on enhancing the quality and coverage of multi-modal knowledge graphs, reducing noise impacts on feature extraction, and developing new algorithms for more effective integration and utilization of different modal data. Moreover, we will explore novel methods to improve the model's sensitivity to dynamic changes in user preferences and its predictive accuracy, as well as to enhance the explainability of the recommendation system.

## 7. Conclusions and Future Work

In this paper, we explored the potential of knowledge graphs in the application of tourist attraction recommendation, particularly by incorporating multi-modal information of attractions as attributes of the attraction entities to enhance the representation of their embedded features. We have designed a new model, MMKG-PAR , which is based on GNNs. This model not only effectively processes and learns the entities and their relationships within the knowledge graph but also aggregates the neighbor information of each entity in a hierarchical recursive manner, while introducing an attention mechanism to distinguish the importance of different entities and relationships. Compared to traditional user embedding strategies, MMKG-PAR adopts an interaction-time-weighted user embedding method, which more accurately reflects the changes in user preferences over time. Furthermore, the model utilizes advanced pre-trained models to extract features of multi-modal information and explores effective strategies for multi-modal integration. Based on real data from the "Travel Yunnan" app, we constructed two collaborative multi-modal knowledge graph datasets and validated the rationality and effectiveness of the MMKG-PAR model through extensive experiments.

The contribution of this research to sustainable tourism lies in its ability to ensure that tourists receive more meaningful and customized information. This not only can enhance the satisfaction level of travel experiences but also indirectly promotes responsible travel behaviors and highlights sustainable tourist attractions, supporting the sustainable development of the tourism industry. This research represents a preliminary exploration in the field of travel recommendation using multi-modal knowledge graphs, laying a foundation for subsequent studies and unveiling numerous intriguing research possibilities. For instance, we plan to further enrich our dataset to encompass comprehensive recommendations

for dining, accommodation, transportation, shopping, and entertainment in the tourism process. In terms of modality, we aim to expand the types of modalities related to tourism entities, allowing for a broader investigation and analysis of user interest preferences. This not only aids in enhancing the attractiveness and competitiveness of tourist destinations but also offers new perspectives and approaches for achieving sustainable development goals. Through this advanced, intelligent recommendation system, we can more accurately meet tourists' needs, thereby fostering a harmonious coexistence between economic growth and environmental protection.

**Author Contributions:** Conceptualization, G.Z. and H.L.; methodology, G.Z.; software, G.Z.; validation, G.Z., S.L. and B.W.; investigation, G.Z. and Z.D.; data curation, G.Z. and Z.D.; writing—original draft preparation, G.Z.; writing—review and editing, S.L., B.W. and H.L.; supervision, H.L; project administration, H.L. All authors have read and agreed to the published version of the manuscript.

**Funding:** Supported by the 2023 Opening Research Fund of Yunnan Key Laboratory of Digital Communications (YNJTKFB-20230686, YNKLDC-KFKT-2023xx).

**Informed Consent Statement:** Informed consent was obtained from all subjects involved in the study.

**Data Availability Statement:** Due to privacy protection reasons, these data are not publicly available. If needed, one can request access to the data used in this study from the corresponding author.

**Conflicts of Interest:** The authors declare no conflicts of interest.

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
