# Peer review of "MMKG-PAR: Multi-Modal Knowledge Graphs-Based Personalized Attraction Recommendation"

_sustainability, doi:10.3390/su16052211_

Round 1

Reviewer 1 Report

Comments and Suggestions for Authors

I believe it is a well-written article highlighting personalized attraction recommendations in the tourism industry. This approach will deeply help to provide targeted tourism products by satisfying visitors’ needs and wants.

Comments to authors

·         Although mentioned in the Conclusions section the issue of sustainable development goals, the authors should elaborate further on these issues from the early start, namely in the Introduction section. They can mention how this approach will advance and promote sustainable or responsible tourism with a long-term perspective.

·         This is a really well-justified research effort. I suggest that the authors should dedicate a few lines regarding their contribution and the potential gap in the relevant literature. This will help the readers understand the study's purpose in depth. Also, the authors can provide a few more literature references to strengthen their scientific arguments for this research.

·         The Authors should provide any potential limitations of their research. For instance, is there any other database that can be used to meet the purpose of the study?

 I propose resubmission of the manuscript based on the above minor revisions. 

Comments on the Quality of English Language

Minor editing on the English Language

Reviewer 2 Report

Comments and Suggestions for Authors

The article addresses an interesting technical problem of how to optimize recommendation algorithms in online marketing. The paper has a logical structure and the authors use correct methods. The content is balanced, relies on relevant sources and provides an interesting insight that can be useful in practice as well as for further academic discussion. However, in order for Sustainability readers to understand the significance of the use case, the authors need to explain in the introduction or in the conclusion part in more detail the link to tourism, destination marketing and in particular to tourism sustainability. It is not clear in the form in which the manuscript was submitted for assessment. For example, advertising messages promoting the territories (places) by the identification of the three types of signs (symbols) contained in the ads: (1) Iconic Signs that can be considered as the representative, respectively the reference or the reference to the territory as a whole. In other words, there is a real similarity between the symbol and the reality because they represent exactly what we see (e.g. the photo of the city or the silhouette of Bratislava Castle as the reference to the capital of Slovakia, the Eiffel Tower as the symbol of Paris); (2) Indexical Signs that indicate the causal relationship or the context to the recipient. The sender is not arbitrarily designated but he is directly (physically or causally) associated with the recipient. By means of this connection, the causal relationship or association is either observed or implied. The city’s registration signs are not the identical image of the city, but their display preserves in some way the connection to the territory they refer to (the derived context). For example, the silhouette of the crown as the reference to the Buckingham Palace, respectively to London as the seat of the sovereign; (3) Symbolic Signs - by means of which the objects can indicate the association of a wide range of activities, mind states or lifestyles. There is no similarity or hint between the city and the symbol that represents it. The fact that we associate some sign with the particular territory is the result of the convention. We cannot decipher the symbolic sign intuitively but we must learn them. For example, we have learned that a red square divided by the white cross into four parts refers to Switzerland (for more see: Matlovičová K., Tirpáková E., Mocák P. (2019). City Brand Image: Semiotic Perspective. A Case Study of Prague. Folia Geographica, Volume 61, No. 1, pp. 120 -142, ISSN 1336-615. In conclusion: the above comments in no way diminish the quality of the study. I would summarise that the study has the potential to appeal to a wide range of professionals once the once the above comments have been included.

I recommend the study for publication after minor revision.

Author Response

Dear Reviewer,

Thank you very much for dedicating your valuable time and effort to review our manuscript and for providing profound and constructive feedback. We highly value your comments and agree that a more detailed discussion on tourism, destination marketing, and particularly the significance of tourism sustainability in our paper would significantly enhance the practical applicability and academic contribution of our research.

Following your suggestions, I have revisited the issue of sustainability goals in the introduction of my paper. Specifically, from lines 23-27 on the first page, the text emphasizes that sustainability is regarded as a core consideration in tourism planning and development. At line 27 on the first page, Reference 1 is introduced, which argues that by integrating advanced artificial intelligence technologies, it is possible to effectively address environmental, economic, and social challenges in the tourism industry, thereby achieving the objective of sustainable tourism. From lines 29-31 on the first page, the section mentions the positive effects of personalized recommendations on the sustainable development of the tourism industry, destination marketing, and the prosperity of local economies. From lines 67-74 on the third page, it cites relevant case studies and literature, such as the research by Matlovičová and others, to further support our argument, elaborating how a rich source of information can enhance the accuracy and reliability of recommendation systems within the tourism sector. From lines 70-84 in the same paragraph, it reiterates that recommendation systems based on multimodal knowledge graphs will aid in the sustainable development of the tourism industry and have a promotional effect on destination marketing, providing an effective promotional strategy for the tourism industry.

We believe that with these adjustments, our research will not only provide valuable insights for practitioners but also stimulate further academic discussions on the role of recommendation systems in promoting sustainable development of tourism destinations.

Yours sincerely,

Reviewer 3 Report

Comments and Suggestions for Authors

Si prega gli autori di potenziare i riferimenti bibliografici soprattutto riguardo all'impatto degli esiti della ricerca sul turismo sostenibile
